# LEARNING THE HIDDEN SET LOCALLY

## ABSTRACT

Learning elements of the hidden set(s), also known as group testing (GT), is a well-established area in which one party tries to discover elements hidden by the other party by asking queries and analyzing feedback. The feedback is a function of the intersection of the query with the hidden set – in our case, it is a classical double-threshold function, which returns $i$ if the intersection is a singleton $i \in [n]$ and "null" otherwise (i.e., when the intersection is empty or of size at least 2). In this work, we introduce a local framework to this problem: each hidden element is an "autonomous" element and can analyze feedback itself, but only for the queries which this element is a part of. The goal is to design a deterministic non-adaptive sequence of queries that allows each non-hidden element to learn about all other hidden agents. We show that, surprisingly, this task requires substantially more queries than the classic group testing – by proving a super-qubic (in terms of the number of hidden elements) lower bound and constructing a specific sequence of slightly longer length. We also extend the results to the model, where agents belong to various clusters and selection must be done in queries avoiding elements from "interfering" clusters. Our algorithms could be generalized to other feedback functions, to adversarial/stochastic fault-prone scenarios and applied to codes.

Keywords: hidden set, group testing, local testing, non-adaptive queries, deterministic algorithms, lower bounds, clusters.

## 1 INTRODUCTION

In the Group Testing (GT) research field, introduced by Dorfman in 1943 Dorfman (1943), the goal is to identify all elements of an unknown set $K$ by asking queries and analyzing answers (so called, feedback vector). Originally GT was applied for identifying infected individuals in large populations using pooled tests, and it has also been very vibrant recently during and after the COVID-19 pandemic Augenblick et al. (2020); Mallapaty et al. (2020); Sinnott-Armstrong et al. (2020). GT has also applications in various areas of Machine Learning, such as: simplifying multi-label classifiers Ubaru et al. (2020), approximating the nearest neighbor Engels et al. (2021), or accelerating forward pass of a deep neural network Liang & Zou (2021). Other applications of GT include stream processing, c.f., extracting the most frequent elements Cormode et al. (2003); Cormode & Muthukrishnan (2005); Cormode & Hadjieleftheriou (2008); Yu et al. (2004); Kowalski & Pajak (2022a), coding Kautz & Singleton (1964); Porat & Rothschild (2011b); Cheraghchi & Ribeiro (2019) and network communication, c.f.: Clementi et al. (2001); Kowalski & Pelc (2003); Jurdzinski et al. (2018). More information, applications and links could be found in the seminal book Du et al. (2000) and recent literature Klonowski et al. (2022); Kowalski & Pajak (2022b).

In this work we consider one of the classical query's feedback models: if the intersection of the query with the hidden set is a single element, the id of this element is returned (we say that this element is selected); otherwise an arbitrary null value is returned. We focus on deterministic non-adaptive solutions, i.e., when a sequence of queries is determined by GT algorithm prior the hidden set is selected by some adversary. The goal of the algorithm, often called a *selector*, is to select every element of the hidden set, using as small number of queries as possible. It is already known that GT in this model can be solved using $O(k^2 \log(n/k))$ queries De Bonis et al. (2003), and an explicit polynomial-time construction of length $O(k^2 \log n)$ exists Porat & Rothschild (2011a). The best known lower bound on the number of queries is $\Omega(\min\{k^2 \log n / \log k, n\})$ Clementi et al. (2001).

In this work, inspired by recent papers by Jurdzinski et al. (2017; 2018) on collision-avoiding communication, we pursue a study of non-adaptive GT enhanced by two additional properties.

First, we require *local learning*, that is, each element in the near universe (but not in the hidden set) needs to learn about the hidden set. However, it can do so only by being included in queries that simultaneously select hidden elements. Note that these two goals, selection and local learning, are not necessarily aligned: selection would rather produce smaller queries, to avoid intersections of size bigger than 1 with the hidden set; quick local learning, on the other hand, would like to place as many near elements to the query as possible – so that many of them could learn, in case the intersection with the hidden set is a singleton. Thus, efficient implementation of selection with simultaneous local learning is a challenging task.

Second, we go further to two-dimensional space[1] and ask question of efficient local learning within any single one-dimensional sub-space, simultaneously *avoiding* any elements from restricted union of any other $\ell$ one-dimensional spaces (here, called clusters). Intuitively, presence of such elements in queries may cause negative inference to the local learning process, e.g., in testing biological/chemical samples, or shared-medium communication (learning "free channels"), or clustering records that are close to being independent. Similarly as in local learning, avoiding any subset of other clusters may not be aligned with simultaneous local learning in a single cluster (the choice of which sub-space is learning and which to avoid could be made arbitrarily by an adversary); therefore, its efficient implementation is challenging.

Apart from providing efficient polynomial-time constructions and almost matching lower bounds on the number of queries in the two considered extended types of group testing, we provide several extensions, remarks and open directions – see details in "Our contribution" and beyond in Section 2.

**Motivation for this work.**

*Non-adaptive vs. adaptive group testing.* There is a distinction between *adaptive* GT, which allows for designing each consecutive queries with knowledge of the results of the preceding queries, and *non-adaptive* GT, which requires that all queries are designed in advance, without any information about results of other queries. While adaptive approach might, under some assumptions, result in more efficient GT schemes – i.e., with a significantly smaller number of queries than in the non-adaptive case – there are at least two advantages of the non-adaptive GT which make it very attractive. Firstly, while adaptive GT schemes can be executed only in a *sequential* way (because each consecutive query may depend on the results of the previous queries), all the queries of the non-adaptive scheme can be executed in *parallel*. This property may, e.g., give much faster testing scenarios in the settings where equipment/environment allows (or make it cheaper) for performing many tests in parallel. Secondly, non-adaptive schemes can be transformed into – and in fact are, according to some definitions, equivalent to – the *encoding/decoding methods* satisfying some specific properties. Well known and very close examples of such application of non-adaptive schemes – both as group testing schemes and as codes – are some types of error correcting codes, e.g., Superimposed Codes.

*Local vs. the "standard" centralized group testing.* Our locality paradigm is designed to support settings in which – because of privacy, security and safety issues or the danger of data leaks – access to the results of particular tests is limited to those entities which are originally included in the test (while staying hidden for the "outside world"). In such types of GT frameworks, the participants outside of the hidden set are allowed to learn only the results of the queries which they belong to. This property is modeled in the definition of the local selector (Section 2). In the clustered case (i.e., local avoiding selector) we address a more sophisticated privacy mechanism, where tests performed within various groups (clusters) are only supposed to allow for reconstruction of the hidden set within those groups. The clustered case can also be used in order to address a scenario where sampled input (some digital data or e.g. some biological material) is supposed to be tested for various properties, under multiple types of data, biological or chemical components, corresponding to clusters. However, some of those properties might be similar or conflicting, such that mixing samples with probable occurrence of at least two of them could give "false positives" for the other.

*Two-thresholds vs. one-threshold feedback with $s = 1$.* For the sake of presentation of our ideas, we focus on a specific variant of a very general two-threshold feedback model with thresholds $s_1 \leq s_2$ such that the result of the query determines whether the size of the intersection of a query set and the hidden set is in the range $[s_1, s_2]$. Our variant assumes $s_1 = s_2 = 1$ and if the intersection size is 1, it also returns the id of the singleton. This feedback function is popular e.g., in designing collision-free

---

[1]The first coordinate could be viewed as an element individual id, while the other – as its cluster id.

wireless communication schedules De Marco & Stachowiak (2017), but there are other popular feedbacks considered in the literature. Probably the most common feedback function assumes one threshold $s$ such that the result of a query is positive iff the size of the intersection of a query set and the hidden set is at least $s$, cf., Dorfman (1943); Klonowski et al. (2022). We have chosen the two-threshold model as it is most convenient to convey ideas and do the formal analysis. However, in Section 5.2 we propose a new efficient transformation between the two abovementioned feedback models – it is different from ones considered in the literature, cf., Kowalski & Pajak (2022b), because it has to take into account locality of decoding process.

**Paper overview.** Section 2 formalizes the model, the problem and describes our contribution. Sections 3 and 4 provide technical details regarding polynomial constructions and lower bounds for local and avoiding selectors. Interesting extensions and corresponding open directions are developed in Section 5. Related work is provided in Section 6. Some proofs, related work on other feedback models and more open problems are deferred to the Appendix, due to space limit.

## 2 MODEL, PROBLEM, CONTRIBUTION AND PRELIMINARIES

Consider the universe of all elements – set $\mathcal{N} = [n] = \{0, \ldots, n-1\}$. Throughout the paper, we will associate an element with its identifier. Let $K$, with $|K| \leq k$, denote a hidden subset of $\mathcal{N}$, chosen arbitrarily by an adversary. It is typically assumed in the literature that $k$ is substantially smaller than $n$, denoted $k \ll n$. Let $\mathcal{Q} = \langle Q_1, \ldots, Q_m \rangle$ be a fixed sequence of $m$ queries generated by a given non-adaptive algorithm. Here, a non-adaptive algorithm is also colloquially called in the literature a $(n, k)$-*selector* or a $(n, k)$-*strong selector* (as the resulting sequence of queries is in fact a fixed mathematical structure), while $m$ is called the *length* or the *size* of the selector.

A general feedback function $\mathcal{F}$ is a function from subsets of $\mathcal{N}$ into an arbitrary domain. A function is applied to $K \cap Q_i$ and the result is called a feedback for query $Q_i$. In our work, we assume a classic feedback $\mathcal{F}$ that returns $null$ in $|K \cap Q_i| \neq 1$ and returns $x$ if $K \cap Q_i = \{x\}$. In the latter case, we say that the query *selects* element $x$ from the hidden set. The goal of the selector is to have every element of the hidden set selected by some query, for any possible hidden set.

In what follows, we present an extension of the classic notion of selector, defined above, by two additional properties that we require from selectors: locality and avoidance. They were introduced recently by Jurdzinski et al. (2018) in the context of collision-avoiding schedules in network communication.[2]

**Local Selectors.** A sequence $\mathcal{Q} = (Q_1, \ldots, Q_m)$ of sets over $[n]$ satisfies *Local Selection* property (or LocS property, for short) for a set $K \subseteq [n]$, if for any $x \in K$ and any $y \notin K$, there is a set $Q_i \in \mathcal{Q}$ such that $K \cap Q_i = \{x\}$ and $y \in Q_i$. One may interpret the above definition as a non-hidden element $y$ being a "witness" of a selection of a hidden element $x$, or alternatively, a non-hidden element $y$ learning that $x$ is in the hidden set $K$ but only if $y$ itself is in the current query $Q_i$. (Note that although $y \in Q_i$, $y$ is not in $Q_i \cap K$ as it is not a hidden element.)

A sequence $\mathcal{S} = (S_1, \ldots, S_m)$ is an $(n, k)$-*Local-Selector* (or $(n, k)$-*LocS*, for short) of length $m$ if, for every subset $K \subseteq [n]$ of size $k$, the family $\mathcal{Q}$ satisfies the LocS property for $K$. The following *non-constructive upper bound* was proved using a probabilistic argument.

**Lemma 1.** *Jurdzinski et al. (2018) For each positive integers $n$ and $k \leq n$, there exists an $(n, k)$-LocS of length $O(k^3 \log n)$.*

One can generalize the notion of $(n, k, \ell)$-LocS even further – to the situation that LocS property must hold only in sub-spaces (with fixed second coordinate, and called clusters) of a two-dimensional space. Even more, that this property holds using only queries that avoid a given set of $\ell$ other clusters. Intuitively, having elements from other clusters in the query may negatively influence, or even clash, the learning process within a given cluster – hence, the goal is to do local learning of $k$ hidden elements within the cluster and simultaneously avoiding the other $\ell$ "bad" clusters. More formal definitions follow.

---

[2]Jurdzinski et al. (2018) called these extended selectors $(n, k)$-witnessed strong selector (or $(n, k)$-wss) for our $(n, k)$-LocS, and $(n, k)$-witnessed clusters aware strong selector (or $(n, k, \ell)$-wcass) for our $(n, k, \ell)$-LocAS. In our work, we propose a unified system of names refined to the GT area: locality and avoidance.

**Local Avoiding Selectors.** We say that a set $Q \subseteq [n]^2$ is *free* of $\phi \in [n]$ if for all $(x, \phi') \in Q$ we have $\phi' \neq \phi$. A set $Q$ is free of a given set $C \subseteq [n]$ if $Q$ is free of each element $\phi \in C$. We call a set of pairs $K \times \{\phi\}$ a *slice*, set $[n] \times \{\phi\}$ a *cluster*, and $\phi$ a *cluster number*. Let $K \subseteq [n] \times \{\phi\}$ be a set of elements that we would like to select locally in the cluster $\phi$, and $L \subseteq [n] \setminus \{\phi\}$ be a set of cluster numbers in conflict with the cluster $\phi$, i.e., elements of these clusters we want to avoid when locally selecting elements in $K$. Then, a sequence $\mathcal{Q} = (Q_1, ..., Q_m)$ of subsets of $[n]^2$ satisfies *Local Avoiding Selection property* (LocAS property, for short) for $K$ with respect to $L$ if for each $x \in K$ and each $y \notin K$ from cluster $\phi$ (i.e., $y \in [n] \times \{\phi\}$) there is a set $Q_i$ such that $Q_i \cap K = \{x\}$, $y \in Q_i$ and $Q_i \cap ([n] \times L) = \emptyset$ (i.e., $Q_i$ is free of clusters from the set $L$ of cluster names). In less formal words, LocAS property requires that for each $x \in K$ and each $y \notin K$ such that $y \in [n] \times \{\phi\}$ is in the same cluster as $x$: $x$ is selected by some $Q_i$, $y$ learns about $x \in K$ (i.e., $y \in Q_i$), and $Q_i$ is free of the clusters from $L$ (i.e., elements from clusters $L$ do not interfere learning by $y$ about $x$).

A sequence $\mathcal{Q} = (Q_1, ..., Q_m)$ of subsets of $[n]^2$ is an $(n, k, \ell)$-*Local-Avoiding-Selector* (or $(n, k, \ell)$-LocAS, for short) if for any set $L \subseteq [n]$ of size $\ell$, any $\phi \notin L$ and any set $K \subseteq [n] \times \{\phi\}$ of size $k$, selector $\mathcal{Q}$ satisfies LocAS property for $K$ with respect to $L$. The following *non-constructive upper bound* was proved using a probabilistic argument.

**Lemma 2.** *Jurdzinski et al. (2018) For each natural $n$, and $k, \ell \leq n$, there exists an $(n, k, \ell)$-LocAS of length $O((k + \ell)\ell k^2 \log n)$.*

**Our technical contribution.** In this work we present *polynomial time constructions* of efficient $(n, k)$-LocS and $(n, k, \ell)$-LocAS. The former has length $O(k^3 \log^2 n(\log_k n + (\log \log n / \log k)^2))$, c.f., Section 3 and Theorem 3.1, while the latter – length $O((k + \ell)\ell k^2 \log^3 n(\log_k n + (\log \log n / \log k)^2)) = O((k + \ell)\ell k^2 \text{ polylog } n)$, c.f., Section 4 and Theorem 4.1. Both our results give almost the same formulas as the best known *existential results* – see the cited Lemmas 1 and 2, respectively. Those results only proved, by using a probabilistic argument, that selectors of similar length exist, without showing how to construct them, c.f., Jurdzinski et al. (2017; 2018). Ours is the first and efficient construction of such selectors.

We complement our constructive results by proving almost-matching lower bounds, correspondingly: $\Omega(\min\{k^3 \log_k n, kn\})$ on the length of any $(n, k)$-LocS, see Section 3 and Theorem 3.2, and $\Omega(\ell \cdot \min\{k^3 \log_k n, kn\})$ on the length of any $(n, k, \ell)$-LocAS, see Section 4 and Theorem 4.2.

Extensions of technical results to fault-prone testing environments, different feedback functions (such as classic beeping) and applications to codes, are given in Section 5.

**A note on inefficiency of a product of selectors.** One could be tempted to construct $(n, k)$-LocS or $(n, k, \ell)$-LocAS by taking a product of two classic $(n, k)$-selectors, or two classic $(n, k)$-selectors and one $(n, \ell)$-selectors, respectively. This way, each of the three properties: selection, locality (witnessing) and avoidance, would be assured "independently" by the means of a different selector in such product. Note, however, that such constructions would not be efficient, and result in linear or even quadratic overhead comparing to our constructions, due to the super-quadratic lower bound on the length of a classic selector – $\Omega(\min\{k^2 \log n / \log k, n\})$ Clementi et al. (2001) (hence, the product will be of order at least $k^4$). Details about the definition of selectors' product and an argument why it satisfies locality are given in Appendix A.1.

**Preliminaries and notation.** A sequence $\mathcal{Q} = (Q_1, ..., Q_m)$ of queries could also be, equivalently, viewed as, and represented by, a 0-1 matrix, in which rows represent elements of the universe, columns represent queries, and an intersection of a row with a column stores value 1 iff the element corresponding to the row belongs to the query corresponding to the column. Such matrix correspond to a *matrix of some specific code*, see Section 5.3 for relation between non-adaptive GT and codes.

We say that $\mathcal{Q}$ is *constructible in polynomial time* if there exists a polynomial-time algorithm, that given parameters $n, k, \ell$ (whichever are relevant) outputs an appropriate sequence of queries satisfying the requirements. W.l.o.g., in order to avoid rounding in the presentation, we assume that $n$ and other crucial parameters used in this work are powers of 2.

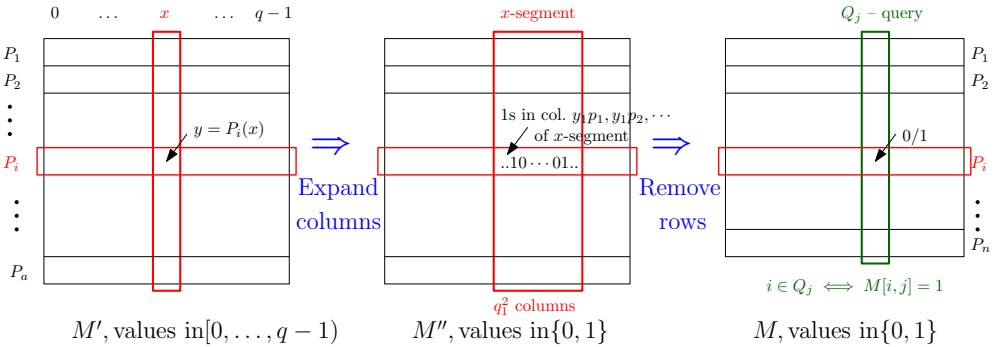

Figure 1: An illustration presenting consecutive matrices $M''$, $M'$ and $M$ used in the algorithm building $(n, k)$-LocS. On the figure, $a = q^d$ for the smallest $d$ such that $q^d > n$, $y_1$ is the $y$th largest prime number which gives $y_1 = O(y \log y)$, $p_1, p_2, \ldots$ denote consecutive primes (i.e., $p_1 = 2$, $p_2 = 3$, $p_3 = 5$ and so on, $y_1 = p_y$ according to this notation).

## 3  LOCAL SELECTORS

The following polynomial-time algorithm produces an $(n, k)$-LocS of length polylogarithmically close to the existential results. Later, we also show that it is actually polylogarithmically close to the absolute lower bound on the lengts of any $(n, k)$-LocS, c.f., Theorem 3.2.

1. Let $d = \lceil \log_k n \rceil$ and let $q = c \cdot k \cdot d$, for some constant $0 < c \le 2$, be a prime number such that $q^{d+1} = (c \cdot k \cdot d)^{d+1} \ge n$. Note that such constant $c$ exists, because $(kd)^{d+1} > n$ and between two integers $\lceil (kd)^{d+1} \rceil$ and $2 \cdot \lceil (kd)^{d+1} \rceil \le \lceil (2kd)^{d+1} \rceil$ there is at least prime number (by the well known distribution property of prime numbers). Let $q_1$ be the $q$th prime number.

2. Consider all polynomials $P_i$ of degree $d$ over field $[q]$, for $1 \le i \le q^{d+1}$. Notice that there are $q^{d+1}$ of such different polynomials.

3. Create a matrix $M'$ of size $q^{d+1} \times q$. Each row $i$ contains subsequent values $P_i(x)$ of polynomial $P_i$ for arguments $x = 0, 1, \ldots, q - 1$, where $x$ is the column number (columns of $M'$ are numbered from 0 to $q - 1$). A matrix $M''$ is created from $M'$ as follows: each value $y = P_i(x)$ is represented and padded in $q_1^2$ consecutive columns of 0s and 1s, where value 1 is on positions $y_1 \cdot z$, for any prime number $z \le q_1$ and for $y_1$ defined as the $y$th prime number; all other positions are filled with value 0. We call these columns an $x$-*segment*. Notice that each row of $M''$ has $q \cdot q_1^2$ columns ($q_1^2$ columns in each segment, where segments correspond to different arguments $x$), thus $M''$ has size $q^{d+1} \times q \cdot q_1^2$; we number the columns from 1 to $q \cdot q_1^2$, where the first segment (corresponding to argument $x = 0$) consists of columns $1, \ldots, q_1^2$, the second segment (corresponding to the argument $x = 1$) has columns $q_1^2 + 1, \ldots, 2q_1^2$, and so on.

4. Remove $q^{d+1} - n$ arbitrary rows from matrix $M''$, creating matrix $M$ with exactly $n$ remaining rows. Recall that we earlier guaranteed that $q^{d+1}$ is at least $n$.

5. Each column of matrix $M$ corresponds to one qyery set $Q_i$ of an $(n, k)$-LocS $\{Q_i\}_{i=1}^{q \cdot q_1^2}$ over the set of $n$ elements, where an element corresponds to a row and it belongs to $Q_i$ iff there is value 1 in the intersection of the corresponding row with $i$th column.

**Theorem 3.1.** *The polynomially constructed family of queries $\{Q_i\}_{i=1}^{q \cdot q_1^2}$ is an $(n, k)$-LocS of length $q \cdot q_1^2 = O(k^3 \log^2 n (\log_k n + (\log \log n / \log k)^2))$, for some suitable constant $0 < c \le 2$.*

*Proof.* Consider a constructed matrix $M$ for some suitable constant $c > 0$. Observe first that two polynomials $P_i$ and $P_j$ of degree at most $d$, for $i \ne j$, could have equal values for at most $d$ different arguments. This is because they have equal values for arguments $x$ for which $P_i(x) - P_j(x) = 0$. However, since $P_i - P_j$ is also a polynomial of degree at most $d$, it could have at most $d$ zeroes. Hence, $P_i(x) = P_j(x)$ for at most $d$ different arguments $x$.

Consider any set of $k$ polynomials $\mathcal{P}$ corresponding to some rows of matrix $M$, and any polynomial $P_j \notin \mathcal{P}$ corresponding to a different row of $M$. Consider any polynomial $P_i \in \mathcal{P}$; clearly, $i \neq j$. Denote by $\mathcal{P}' = \mathcal{P} \setminus \{P_i\}$ the set of polynomials in $\mathcal{P}$ different from $P_i$; they are also different from $P_j$, by the choice of $P_j$. There are at most $(k-1) \cdot d$ different arguments $x$ where some of the $k-1$ polynomials in $\mathcal{P}'$ could be equal to $P_i$, and similarly, at most $(k-1) \cdot d$ different arguments $x$ where some of the $k-1$ polynomials in $\mathcal{P}'$ could be equal to $P_j$. Thus, for $q - 2(k-1) \cdot d$ different arguments, the values of the polynomials $P_i$ and $P_j$ are simultaneously different from the values of polynomials in $\mathcal{P}'$. Therefore, there are at least $q - 2(k-1) \cdot d \geq 1$ segments (corresponding to some arguments) $x$ in which $P_i(x)$ and $P_j(x)$ are different from any $P_{i'}(x)$, where $P_{i'} \in \mathcal{P}'$. Let us pick arbitrarily such a segment/argument $x$. Let $y = P_i(x)$ and $y' = P_j(x)$. Further, let $y_1$ be the $y$th prime number and $y_1'$ be the $y'$th prime number.

Consider column $y_1 \cdot y_1'$ in segment $x$ of matrix $M$. By definition of distribution of 1's in a segment (see point 3 of the algorithm creating matrix $M$), there are values 1 in this column in the rows corresponding to both polynomials $P_i$ and $P_j$. Consider any polynomial $P_{i'} \in \mathcal{P}'$. By the choice of $x$, the value $y^* = P_{i'}(x)$ is different from $y, y'$. Hence, the row corresponding to polynomial $P_{i'}$ has values 1 only at positions $\hat{y} \cdot y_1^*$, where $y_1^*$ is the $y^*$th prime number and $\hat{y}$ is any prime number not bigger than $q_1$. These positions are, however, all different from position $y_1 \cdot y_1'$ in segment $x$, in which both rows corresponding to polynomials $P_i$ and $P_j$ have value 1. This is because at least one prime number in the multiplicative representations of these positions is different from $y_1, y_1'$. It means that this column $x \cdot q_1^2 + y_1 \cdot y_1'$ has value 1 in rows corresponding to polynomials $P_i$ and $P_j$, and value 0 in all rows corresponding to polynomials in $\mathcal{P}'$. This completes the proof that the matrix $M$ is an $(n, k)$-LocS.

The length of the constructed $(n, k)$-LocS is $q \cdot q_1^2 = O(q^3 \log^2 q)$, as $q_1 = O(q \log q)$ by the prime number theorem applied to the largest of the considered prime numbers – the $q$th prime number. We have $q = O(k \log n / \log k)$, and consequently $\log q = O(\log k + \log \log n)$. Thus, the length of the $(n, k)$-LocS is $q \cdot q_1^2 = O(k^3 \log^2 n (\log_k n + (\log \log n / \log k)^2))$. The construction is clearly polynomial, as it is done by enlisting consecutive elements of the algebraic field containing at most $O(n)$ elements, computing values of polynomials (there is a polynomial number of considered polynomials) and padding polynomially many columns modulo prime numbers in $O(n \log n)$. $\qquad\square$

Now we prove a lower bound on the length of any $(n, k)$-LocS, almost matching our constructive result in Theorem 3.1.

**Theorem 3.2.** *Every $(n, k)$-LocS has length $\Omega(\min\{k^3 \log_k n, kn\})$.*

*Proof.* First, observe that the number of 1s in each row $i$ of any $(n, k)$-LocS is $c \cdot \min\{k^2 \log_k n, n\}$, for some constant $c > 0$, because the corresponding columns must form an $(n, k)$-strongly-selective family for which $i$ is a witness; the lower bound on the number of columns in any $(n, k)$-strongly-selective family is $\Omega(\min\{k^2 \log_k n, n\})$, by Clementi et al. (2001).

Let $m$ denote the minimum length of any $(n, k)$-LocS. Consider any row $i$ and remove from the $(n, k)$-LocS all the columns having 1 in row $i$, and then the row $i$ itself. What remains, must be an $(n-1, k-1)$-LocS, as the original $(n, k)$-LocS has to handle, in particular, all subsets of $[n]$ of size $k-1$ taken from $[n] \setminus \{i\}$ with added element $i$, and has length at most $m - c \cdot \min\{k^2 \log_{k-1} n, n\}$. Repeating it recursively, after $j$ steps we get $(n-j, k-j)$-LocS of length at most $m - j \cdot c \cdot \min\{(k/2)^2 \log_{k-j} n, n\}$, for $1 \leq j \leq k/2$, and consequently for $j = k/2$: we get $(n-k/2, k/2)$-LocS of length at most $m - (k/2) \cdot c \cdot \min\{(k/2)^2 \log_{k/2} n, n\}$. It follows that the length $m - (k/2) \cdot c \cdot \min\{(k/2)^2 \log_{k/2} n, n\}$ must be positive, thus, together with the fact that $\log_{k/2} n = \Theta(\log_k n)$, we get $m > (k/2) \cdot c \cdot \min\{(k/2)^2 \log_{k/2} n, n\} = \Omega(\min\{k^3 \log_k n, kn\})$. $\qquad\square$

# 4 LOCAL AVOIDING SELECTORS

Suppose that we are given a set $X \subseteq [n]^2$ of size at most $k \cdot \ell$ and such that it consists of at most $\ell$ slices of size at most $k$ each. The $(n, k)$-LocS from Section 3 guarantees that all other agents learn set $X$ in $O((k\ell)^3 \text{ polylog } n)$ rounds. Is faster learning possible if we require that only other agents in each slice's cluster learn the slice? Even more, if we additionally would like to avoid clashes between in such learning rounds from other slices of $X$?

The following polynomial-time algorithm produces an $(n, k, \ell)$-LocAS of length polylogarithmically close to the existential result – see Figure 2 for an illustration.

1. Let $d_k = \lceil \log_k n \rceil$, and let $q_k = c \cdot k \cdot d_k$ be a prime number such that $q_k^{d_k+1} = (c \cdot k \cdot d_k)^{d_k+1} \geq n$, for some constant $4 < c \leq 8$. Note that such constant $c$ exists, because $(kd_k)^{d_k+1} > n$ and between two integers: $\lceil (4kd_k)^{d_k+1} \rceil$ and its double $2\lceil (4kd_k)^{d_k+1} \rceil \leq \lceil (8kd_k)^{d_k+1} \rceil$, there is at least one prime number. Analogously, we define $d_\ell$ and $q_\ell$.

2. Consider all polynomials $P_i$ of degree $d_k$ over field $[q_k]$, for $1 \leq i \leq q_k^{d_k+1}$. Notice that there are $q_k^{d_k+1}$ of such different polynomials. Analogously, consider all polynomials $R_i$ of degree $d_\ell$ over field $[q_\ell]$, for $1 \leq i \leq q_\ell^{d_\ell+1}$. Notice that there are $q_\ell^{d_\ell+1}$ of such different polynomials.

3. Create a matrix $M_k'$ of size $q_k^{d_k+1} \times q_k$. Each row $i$ contains subsequent values $P_i(z)$ of polynomial $P_i$ for arguments $z = 0, 1, \ldots, q_k - 1$, where $z$ is the column number (columns of $M_k'$ are numbered from 0 to $q_k - 1$). We trim matrix $M_k'$ to $n$ rows by removing $q_k^{d_k+1} - n$ arbitrary rows.

   Analogously, we create a matrix $M_\ell'$ of size $q_\ell^{d_\ell+1} \times q_\ell$, in which each row $i$ contains subsequent values $R_i(z)$ of polynomial $R_i$ for arguments $z = 0, 1, \ldots, q_\ell - 1$, where $z$ is the column number (columns of $M_\ell'$ are numbered from 0 to $q_\ell - 1$). We trim matrix $M_\ell'$ to $n$ rows by removing $q_\ell^{d_\ell+1} - n$ arbitrary rows.

4. Matrix $M'$ is created from $M_k', M_\ell'$ as follows: there are $n^2$ rows and $q = 3 \max\{q_k, q_\ell\}$ columns. In each row $(i, \phi)$, corresponding to the pair of polynomials $P_i, R_\phi$, and each column $z \in [q]$, we put in the intersection a pair of values $(P_i(z \mod q_k), R_\phi(z \mod q_\ell))$.

5. Matrix $M$ is created from $M'$ as follows: each pair of values $(i^\star, \phi^\star) = (P_i(z \mod q_k), R_\phi(z \mod q_\ell))$ in column $z$ is represented and padded in $(q_k')^2 q_\ell'$ consecutive columns of 0s and 1s as follows. Let $q_k', q_\ell'$ be the prime numbers of order $2q_k$ and $2q_\ell + 1$, respectively (i.e., the $(2q_k)$-th prime number in the order of all prime numbers, and $(2q_\ell + 1)$-st prime number in the order of all prime numbers). Let $p_i$ be the prime number of order $2i^\star$ and $p'$ be any prime number of order $2, 4, 6, \ldots, 2q_k$. Let $p_{\phi^\star}$ be the prime number of order $2\phi^\star + 1$. We put values 1 in columns $p_i \cdot p' \cdot p_{\phi^\star}$, and values 0 in the remaining columns. We call these columns a $z$-*segment*. Notice that each row of $M$ has $q \cdot (q_k')^2 q_\ell'$ columns ($(q_k')^2 q_\ell'$ columns in each segment, where segments corresponds to different columns $z$ of $M'$). Thus, $M$ is an $n^2 \times q \cdot (q_k')^2 q_\ell'$ matrix; we number its columns from 1 to $q \cdot (q_k')^2 q_\ell'$, where the first segment (corresponding to argument $z = 0$) consists of columns $1, \ldots, (q_k')^2 q_\ell'$, the second segment (corresponding to the argument $z = 1$) $(q_k')^2 q_\ell' + 1, \ldots, 2(q_k')^2 q_\ell'$, and so on.

6. Each column of matrix $M$ corresponds to one query set $Q_i$ of an $(n, k, \ell)$-LocAS $\{Q_i\}_{i=1}^{q \cdot (q_k')^2 q_\ell'}$ over the set of $n^2$ elements (corresponding to the rows of $M$).

**Theorem 4.1.** *The polynomially constructed* $\{Q_i\}_{i=1}^{q \cdot (q_k')^2 q_\ell'}$ *is an* $(n, k, \ell)$-*LocAS of length*

$$O\left((k + \ell)k^2\ell \cdot \frac{\log^4 n}{\log^2 k \log \ell \log(k\ell)} (\log k + \log\log n)^2 (\log \ell + \log\log n)\right),$$

*for some suitable constant* $4 < c \leq 8$.

The proof of Theorem 4.1 is deferred to Appendix A.2.

Below we show a lower bound that nearly matches the constructive result analyzed in Theorem 4.1.

**Theorem 4.2.** *Every* $(n, k, \ell)$-*LocAS has length* $\Omega(\ell \cdot \min\{k^3 \log_k n, kn\})$.

*Proof.* Consider $(n, k, \ell)$-LocAS of length $m$. Consider a set $L \subseteq [n]$ of size $\ell + 1$. For any $i \in [n]$, consider a set of pairs $\{(i, j) : i \in [n]\}$, that is, a set of rows in the $(n, k, \ell)$-LocAS corresponding/labeled to/by these pairs. The number of columns that

- have at least one 1 in these rows (i.e., take part in the local selection of any set of size $k$ of pairs with $i$ in their second coordinate), and

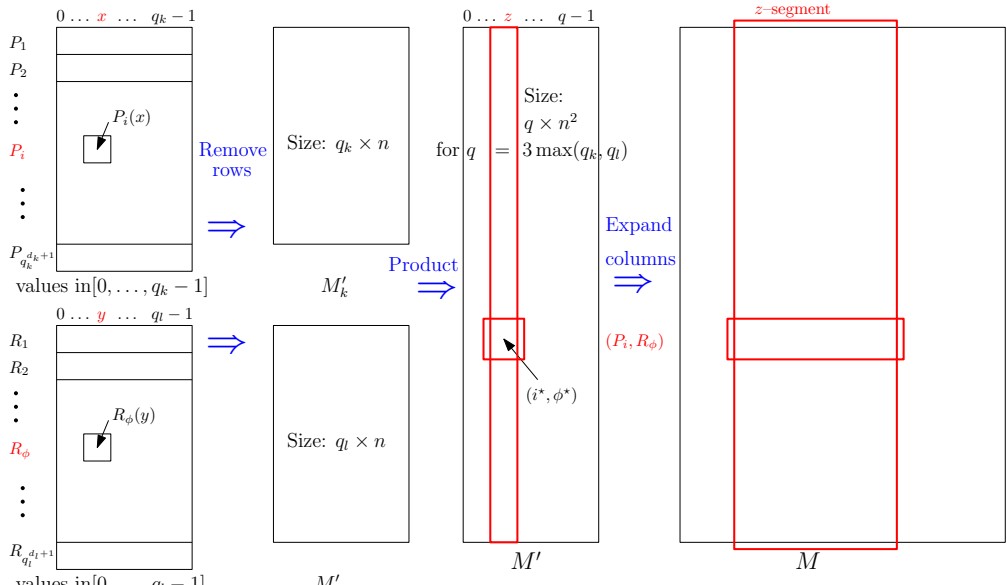

Figure 2: An illustration of consecutive matrices used in the algorithm building $(n, k, l)$-LocAS. On the figure, $(i^\star, \phi^\star) = (P_i(z \bmod q_k), Q_\phi(z \bmod q_l))$, the matrix $M$ is obtained by the replacement of each element of $M'$ with a binary sequence of the length $(q'_k)^2 q'_l$ as described in the item 5 of the description of the algorithm before the theorem.

- do not have any 1 from any row labeled by an element in $L \setminus \{i\}$ (i.e., avoids other elements in $L$ on the second coordinate)

is at least $\Omega(\min\{k^3 \log_k n, kn\})$, by the fact that they must be an $(n, k)$-LocS, to which Theorem 3.2 applies. Denote the set of such columns $C_i$.

Next, if we consider analogous set $C_{i'}$ of columns defined for an element $i' \in L$ different from $L$, it is disjoint with $C_i$, due to the second bullet in the definition of set $C_i$ above (avoidance property). Hence, the total number of columns in the $(n, k, \ell)$-LocAS is

$$|L| \cdot \Omega(\min\{k^3 \log_k n, kn\}) \geq \Omega(\ell \cdot \min\{k^3 \log_k n, kn\}) . \qquad \square$$

## 5 EXTENSIONS

This section presents interesting extensions to technical results obtained in the previous sections, as well as corresponding future directions. Other perspective open problems are given in Appendix B.

### 5.1 FAULT-TOLERANCE OF LOCAL GT

Suppose one would like to be able to decode the hidden set correctly even if some $\alpha$ positions in the feedback vector would be altered by a *worst-case* adversary. More precisely, assume that the adversary could change a feedback from specific id to zero (but cannot produce/forge an id as feedback). Observe that if we use a larger constant $c$ in the constructions, for instance, $c \geq 2 + \frac{\alpha}{kd}$, there will be always some column with correct feedback because the number of "witnessed" segments (and thus, also columns) for a pair $i, j$, which is $q - 2(k-1) \cdot d$, deducted by the number of adversarially changed ones, $\alpha$, is still at least 1 (see the proofs of Theorems 3.1 and 4.1). Increasing constant $c$ obviously increases the lengths of selectors by factor $\lceil \frac{\alpha}{kd} \rceil$. Improving the number of tolerated faults or generalizing to other types of adversarial or stochastic failures is an interesting area for further investigation.

## 5.2 OTHER FEEDBACK FUNCTIONS

We show an example how to convert the obtained results to another popular GT feedback functions – the classic one-threshold setting in which the feedback vector is a 0-1 vector where the value on a position $z$ is equal 1 if the intersection of the $z$-th query with the hidden set is non-empty, and is equal to 0 otherwise. It has also been named beeping feedback in recent literature.

The method is similar to the one guarantying fault-tolerance. If we extend parameter $c$ to be a constant slightly bigger than 2, the asymptotic length and analysis of our constructions become intact. However, the number of "witnessed" columns for any pair $i, j$, which is $q - 2(k-1) \cdot d$, could be made bigger than $d$. Each such column corresponds to a different segment, and thus – to different arguments for which polynomials are evaluated. Obviously, the feedback now is only 1 in such columns, however, $j$ could create as many equations as the number of such columns in order to find the polynomial of any element $i$ in the hidden set. Now, since the number of such equations is at least $d + 1$, the polynomial for $i$ could be interpolated successfully, as its degree is at most $d$. Hence, each element of the hidden set could be successfully found.

Note here that lower bounds from Sections 3 and 4 hold automatically for the beeping feedback, because the previously considered feedback function is richer than the beeping feedback.

Extending our constructions and lower bounds to other types of feedback function, considered in the literature (cf., Klonowski et al. (2022)) is another interesting research direction.

## 5.3 LOCAL GT AS CODES

By definition of GT, one should be able to decode a hidden set from the feedback vector – recall that each position of this vector has been created by applying a given feedback function to the intersection of the query corresponding to this position (queries are designed by the "coding" algorithm) and the hidden set (an arbitrary set, fixed by the adversary). For instance, in case of the beeping feedback function, the feedback vector is computed by applying bitwise OR on the vectors of participating elements, and if another element want to get the feedback locally, it applies bitwise AND to its own vector and the feedback vector. Applying similar methodology to other feedback function which are symmetric Boolean function could result in interesting results in the area of codes, in particular, when the function is applied to the participating codewords and the feedback is decoded locally with avoidance of codewords from different clusters/groups.

## 6 RELATED WORK

*Adaptive* version of the group testing model considered in our work has also been extensively studied, including early work by Capetanakis (1979a;b) and Hayes (1978), who independently found an adaptive, deterministic tree-based algorithm with $O(k + k \log(n/k))$ queries, but using a slightly richer feedback. Greenberg & Winograd (1985) proved a lower bound $\Omega\left(\frac{k \log n}{\log k}\right)$ in this setting. Kowalski (2005) gave an adaptive construction of $O(k + k \log \frac{n}{k})$ queries with the feedback exactly as considered in this work (although, when it comes to polynomial time construction, a polylogarithmic overhead is incurred). The corresponding lower bound was given by Clementi et al. (2001).

Different types of selectors have been widely used to avoid simultaneous access to communication channel or other resources, c.f., Chlebus et al. (2000); Chrobak et al. (2002); Jurdzinski & Kowalski (2012). The witnessed strong selectors, corresponding to our local selectors, were introduced in context of token traversal in an SINR ad hoc network Jurdzinski et al. (2017), while their generalization, called witnessed cluster aware strong selectors (corresponding to our local avoiding selectors) were introduced in Jurdzinski et al. (2018). As mentioned in Section 2, only existential upper bounds on sizes of such selectors have been known until now.

Related work on other GT models can be found in Appendix A.3.

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

## A  APPENDIX

### A.1  TECHNICAL DETAILS ABOUT SELECTORS' PRODUCT

In Section 2 we discussed, in a "Note ...", an alternative approach to constructing local selectors via selectors' product, and why it is much less efficient than the technical constructions in our work. More specifically, one could be tempted to construct $(n, k)$-LocS or $(n, k, \ell)$-LocAS by taking a product of two classic $(n, k)$-selectors, or two classic $(n, k)$-selectors and one $(n, \ell)$-selectors, respectively. This way, each of the three properties: selection, locality (witnessing) and avoidance, would be assured "independently" by the means of a different selector in such product. Note, however, that such constructions would not be efficient, and result in linear or even quadratic overhead comparing

to our constructions, due to the super-quadratic lower bound on the length of a classic selector – $\Omega(\min\{k^2 \log n / \log k, n\})$ Clementi et al. (2001) (hence, the product will be of order at least $k^4$).

To convey more details, first we need to formally define the meaning of a product of two selectors. (Here we only focus on the first paradigm of locality.) It could be seen as a product with "or" operation on pairs of columns. Consider a column $Q_i$ of the first selector and a column $Q'_j$ of the second selector that has length $m'$. In the product, the column $(i-1)m' + j$ is defined as follows: the value stored in a row $v \in [n]$ is 1 if $Q_i[v] = 1$ or $Q'_j[v] = 1$, otherwise it is 0. Having a product of two $(n, k)$-selectors, consider any set $K$ of size $k$, any element $v \in K$ and any $w \in [n] \setminus K$. Consider a column $Q_i$ in the first selector in which $v$ is selected from set $K$ – its existence is guaranteed by the definition of $(n, k)$-selector. If $w \in Q_i$ we are done – there is a selection of $v$ witnessed by $w$. Suppose then that $w \notin Q_i$. Consider set $K' = (K \setminus \{v\}) \cup \{w\}$. It has $k$ elements, thus the second $(n, k)$-selector has a query $Q'_j$ in which $w$ is selected from $K'$. It means that in the product of the two selectors, in the query/column $(i-1)m' + j$, row $v$ has value 1 (by the fact that $Q_i[v] = 1$ and definition of selectors' product), row $w$ has value 1, while all elements in $K \setminus \{v\} = K' \setminus \{w\}$ have value 0. Thus, this column guarantees selection of $v$ from $K$ with $w$ being a "witness".

## A.2 Proof of Theorem 4.1

Consider a constructed matrix $M$ for some suitable constant $c > 0$. The same argument as in the proof of Theorem 3.1 shows that any two polynomials $P$ and $P'$ of degree at most $d$, for any $d$, could have equal values for at most $d$ different arguments.

Consider any set $K$ of $k$ rows of matrix $M$ with the same second coordinate, say $\phi$. Pick an arbitrary row $(i, \phi) \in K$ and row $(j, \phi) \notin K$. Let $P_i, P_j$ be the corresponding polynomials based on the first coordinates, and $R_\phi$ their common polynomial corresponding to their second coordinate $\phi$ (i.e., cluster number). Let $\mathcal{P}$ be the set of all polynomials corresponding to the first coordinates of rows in $K$, and let $\mathcal{P}' = \mathcal{P} \setminus \{P_i\}$. Consider any set $L$ of $\ell$ numbers in $[n] \setminus \{\phi\}$ and let $\mathcal{R} = \{R_v : v \in L\}$.

We first argue that there is a segment $z \in [q]$ such that

(a) $P_i(z) \neq P_{i'}(z)$ and $P_j(z) \neq P_{i'}(z)$, for any $(i', \phi) \in K$ different from $(i, \phi)$, and

(b) $R_\phi(z) \neq R_v(z)$, for every $v \in L$.

There are at most $(k-1) \cdot d_k$ different arguments $z \in [q_k]$ where some of the $k-1$ polynomials in $\mathcal{P}'$ could be equal to $P_i$, and similarly, at most $(k-1) \cdot d_k$ different arguments $z \in [q_k]$ where some of the $k-1$ polynomials in $\mathcal{P}'$ could be equal to $P_j$. Let us call these logical events for an argument $z$ – Event_1 and Event_2, respectively. Note that now we only consider $z \in [q_k]$, as polynomials $P$ are defined over the field of arguments in $[q_k]$. Therefore, the upper bounds on the number of arguments $z \in [q]$ are $(k-1)d_k \cdot \lceil \frac{q}{q_k} \rceil$ for each of these two logical events, which sum up to at most $2(k-1)d_k \cdot \lceil \frac{q}{q_k} \rceil$ arguments $z \in [q]$ for which at least one of these events applies.

Analogously, there are at most $\ell \cdot d_\ell$ different arguments $z \in [q_\ell]$ where some of the $\ell$ polynomials in $\mathcal{R}$ could be equal to $R_\phi$. Let us call such event for argument $z$ – Event_3. Note that here we only consider $z \in [q_\ell]$, as polynomials $R$ are defined over the field of arguments in $[q_\ell]$. Hence, the number of arguments $z \in [q]$ for which it may happen is at most $\ell \cdot d_\ell \cdot \lceil \frac{q}{q_\ell} \rceil$.

To summarize the two previous paragraphs, none of Event_1, Event_2 and Event_3 happens for

$$q - 2(k-1)d_k \cdot \lceil \frac{q}{q_k} \rceil - \ell \cdot d_\ell \cdot \lceil \frac{q}{q_\ell} \rceil \geq q - 2(k-1)d_k \cdot \left( \frac{q}{q_k} + 1 \right) - \ell \cdot d_\ell \cdot \left( \frac{q}{q_\ell} + 1 \right)$$

$$q - 2\frac{q}{c} + 2\frac{q}{ck} - 2\frac{q_k}{c} - \frac{q}{c} + \frac{q}{c\ell} - \frac{q_\ell}{c} \geq q - 4\frac{q}{c} + 2\frac{q}{ck} + \frac{q}{c\ell} \geq 1$$

arguments $z \in [q]$, since $q > 2q_k + q_\ell$ and $c > 4$.

Consider such an argument $z$ and its corresponding segment in the constructed matrix $M$. Consider column $p_i \cdot p_j \cdot p_\phi$ in this segment, where $p_i, p_j$ are the prime numbers of order $2P_i(z \mod q_k), 2P_j(z \mod q_k)$, respectively, and $p_\phi$ is the prime numbers of order $2R_\phi(z) + 1$. By definition, this column of segment $z$ has 1 in rows $(i, \phi)$ and $(j, \phi)$, but not in any other rows $(i', \phi) \in K \setminus \{i\}$, by Event_1 and Event_2. This column does not have a 1 in rows with second coordinate in set $L$, as such rows are not divisible by $p_\phi$ by Event_3. This completes the proof that the matrix $M$ is an $(n, k, \ell)$-LocAS.

The length of the constructed $(n, k, \ell)$-LocAS is $q \cdot (q'_k)^2 q'_\ell = O((q_k + q_\ell)q_k^2 q_\ell \log^2 q_k \log q_\ell)$, as $q'_k = O(q_k \log q_k)$ and $q'_\ell = O(q_\ell \log q_\ell)$ by the prime number theorem applied to the largest of the considered prime numbers – the prime numbers of order $2q_k$ and $2q_\ell + 1$, respectively. We have $q_k = O(k \log n / \log k)$ and $q_\ell = O(\ell \log n / \log \ell)$. Consequently, $\log q_k = O(\log k + \log \log n)$ and $\log q_\ell = O(\log \ell + \log \log n)$. Thus, the length of the $(n, k, \ell)$-LocAS is

$$q \cdot (q'_k)^2 q'_\ell = O((q_k + q_\ell)q_k^2 q_\ell \log^2 q_k \log q_\ell)$$

$$\leq O\left((k + \ell)k^2\ell \cdot \frac{\log^4 n}{\log^2 k \log \ell \log(k\ell)}(\log k + \log \log n)^2(\log \ell + \log \log n)\right) .$$

The construction is clearly polynomial, as it is done by enlisting consecutive elements of the algebraic field containing at most $O(n)$ elements, computing values of polynomials (there is a polynomial number of considered polynomials) and padding polynomially many columns modulo prime numbers in $O(n \log n)$.

### A.3 RELATED WORK ON OTHER MODELS OF GROUP TESTING

In the simplest group testing model, considered in most of the literature Du et al. (2000), the feedback informs only if the intersection between query $Q$ and the hidden set $K$ is empty or not (sometimes it is also called a *beeping model*).

In the quantitative group testing model, also called a coin weighting problem, we are given a set of $n$ coins of two distinct weights $w_0$ (true coin) and $w_1$ (counterfeit coin), out of which up to $k$ are counterfeit ones. The queries correspond to weighing any subset of coins on a spring scale. The feedback, therefore, gives exact number of counterfeit coins in the subset/query. This problem can be solved using $O(k \log(n/k)/\log k)$ non-adaptive queries, c.f., Grebinski & Kucherov (2000). It matches a standard information-theoretic lower bound of $\Omega(k \log(n/k)/\log k)$. Bshouty (2009) considered the problem of efficient polynomial-time constructions of $O(k \log(n/k)/\log k)$ queries that allows to decode the counterfeit coins from the feedback, but only adaptive solution was obtained. Efficient polynomial-time construction of non-adaptive queries remains an open problem. – the best known construction of non-adaptive queries was given recently by Kowalski & Pajak (2022b), however it has a polylogarithmic overhead on the number of queries.

Threshold Group Testing is another feedback model, including a set of thresholds, c.f., Damaschke (2005); De Marco et al. (2020). The feedback returns whether or not the size of the intersection is larger or smaller than each threshold. De Marco et al. (2021) showed that it is possible to define an interval of $\sqrt{k \log k}$ thresholds resulting in an algorithm with $O(k \log(n/k)/\log k)$ queries.

If a feedback only returns whether the size of the intersection $|Q \cap K|$ is odd or even, then $O(k \log \frac{n}{k})$ queries are sufficient to decode the hidden set, as shown by Censor-Hillel et al. (2015).

Group testing with general feedback functions was studied and put together in Klonowski et al. (2022); Kowalski & Pajak (2022b) – more specifically, how the complexity of the feedback function influences learning time.

Randomized solutions to group testing have also been widely studied – the reader may find hot topics and references in the recent papers by Gebhard et al. (2019); Coja-Oghlan et al. (2020); Feige & Lellouche (2020); Bay et al. (2020). We would like to point out that randomized solutions are typically designed for an oblivious adversary, of weaker power than the unbounded adversary considered in the deterministic setting (including our work). Moreover, randomized queries requires a substantial number of truly random bits, in order to avoid biases – detrministic solutions are fair in the sense that they reveal all hidden elements.

## B DISCUSSION AND OTHER FUTURE DIRECTIONS

This work introduced to the GT area new concepts (translated from network communication theory) of local learning and avoidance of elements from different clusters. We designed polynomially constructable and nearly optimal, in terms of the number of queries, non-adaptive query systems that observe both new properties. To justify near-optimality of our constructions, we also proved the corresponding lower bounds.

Apart of open problems already suggested in Section 5, there are more interesting directions to follow.

First and most straightforward future direction is to improve the remaining gaps on the lengths of LocS and LocAS selectors, since the constructions and corresponding lower bounds are not strictly matching, as well as analyzing their other properties, e.g., query sizes.

Second and main future direction regards direct generalization of the proposed concepts into multi-dimensional system, and further into more structured systems (e.g., graphs, hypergraphs, matroids, polytopes, etc.).

Third open question addresses robustness of the proposed concepts and their implementations. For instance, how to express privacy in terms of the proposed or related concepts of locality and avoidance?

Finally, as a fourth topic, one could ask whether randomization substantially helps in local learning, especially against an adaptive adversary who may dynamically tailor the hidden set (as long as it is compatible with the obtained feedback at any time)? If the answer is yes, what is the minimum amount of randomness (entropy) needed and how it affects learning time?

