# OpenReview forum: "Learning the Hidden Set Locally"
_ICLR.cc/2024/Conference — Submitted to ICLR 2024_

### Official Review · Reviewer_2oQv · 2023-10-14

**Soundness:** 3 good
**Presentation:** 3 good
**Contribution:** 3 good
**Rating:** 6
**Confidence:** 3

**Summary:**

This paper studies a variant of the group testing problem, where within a universe $N$ of $n$ elements there is an unknown “hidden” set $K$ of $k$ elements, and the goal is to recover the hidden set using queries. In the variant studied here, a query $Q$
 is simply a subset of the universe, and it returns the following. If $|Q \cap K| = 1$ then the (single) element at the intersection is being returned. Otherwise, nothing is returned. (Notably, the most common variant of group testing is one where if $|Q \cap K| \geq 1$ then “yes” is returned, and otherwise “no”.)

The authors consider two novel variant of interest under the above query model. The first variant is a so called “local” one, where the goal is that for each element $x \in N \setminus K$ (i.e., each element of the universe not from the chosen set), and each $y \in K$, there will be at least one query containing both $x$ and $y$ (and no other element from $K$, as per our query model). The second model is a local one with an added constraint, where we want the above query to not only contain $x$ and $y$, but also be free of a pre-determined (but unknown) forbidden set of elements.

The main result is an explicit construction of a solution to these group testing problems for both of the above models. Both results have query complexity cubic in $k$, which is shown to be tight up to lower order terms, and for the second, “local avoiding” model the dependence in $\ell$ is polynomial (but the results are not tight). Technically, the construction is algebraic and seems to rely on properties of polynomial of low degree over finite fields. The lower bound follows by a simple recursive interpretation of the definition of the local model. One interesting artifact of the results is a separation between these the query complexity of these local models and the complexity of the classical model, where we only need to uncover the hidden set “centrally”.

**Strengths:**

1. Novelty: Presenting new local models for group testing.
2. Elegant and strong/near-tight constructive results for the newly presented models.
3. A well written paper. I have reviewed a previous version of this paper, and the writing in the current version is substantially improved, with clearer motivation, better presentation of the model, and more robust proofs.

**Weaknesses:**

1. Scope: Not clear that this paper will be of interest to a wide ML audience. Looking at the references, maybe a distributed computing venue would be more fitting.
2. Is the model interesting? I am not completely convinced, for example it is hard to imagine privacy constraints (as suggested by the authors) forcing the intersection size to be exactly one.

**Questions:**

None -- my questions were addressed when reviewing this paper in the past.

---

> ### Author Response · Authors · 2023-11-23
>
> We thank the Reviewer for noticing the improvements we  made since the previous submission of this work. We hope that our answers below to the weaknesses of the submitted version, pointed out in the review, could convince the Reviewer to support our paper.
>
> **Addressing weaknesses:**
>
> 1. **Scope: Not clear that this paper will be of interest to a wide ML audience. Looking at the references, maybe a distributed computing venue would be more fitting.**
>
> In Section 1 ("Introduction"), we give references to papers describing and exploring applications, in particular, in:
>
> + Machine Learning, such as: simplifying
> multi-label classifiers, cf. Ubaru et al. (NeurIPS 2020), approximating the nearest neighbor, cf. Engels et al. (NeurIPS 2021),
> or accelerating forward pass of a deep neural network, cf. Liang \& Zou (ISIT 2021);
>
> + stream processing, e.g., extracting the most frequent elements, see Cormode et al. (VLDB 2003),
> Cormode \& Muthukrishnan (ACM ToDS 2005), Cormode \& Hadjieleftheriou (VLDB 2008), Yu et al. (VLDB 2004), Kowalski
> \& Pajak (IJCAI 2022);
>
> + coding, cf. Kautz \& Singleton (IEEE ToIT 1964); Porat \& Rothschild (IEEE ToIT 2011); Cheraghchi \&
> Ribeiro (ISIT 2019).
>
> Many of those papers, and other related ones, got accepted by ML and AI community (NeurIPS, IJCAI), as well as related information and coding theory (ISIT, IEEE Trans. on Information Theory ToIT) and databases (VLDB, ACM Trans. on Database Systems ToDS).
> A substantial part of the whole book cited in our submission (Dingzhu Du, Frank K Hwang, and Frank Hwang. Combinatorial group testing and its applications, volume 12. World Scientific, 2000) is devoted to applications of group testing paradigm to data science and learning. See also examples in Kowalski \& Pajak paper in NeurIPS 2022.
>
> Specific applications close to our exact settings are discussed in more detail in Sub-section ''Motivation for this work.'', including various variants of the problem.
>
> On the other hand, there is also a rich literature on group testing applications, which we did not include in our work because it studies
> slightly different GT settings. For instance, average case instead of worst case performance over some probabilistic distribution of the sets of infected items, random choice of tests) with applications in information theory (see e.g., A. E. Alaoui, A. Ramdas, F. Krzakala, L. Zdeborova, and M. I. Jordan; Decoding from Pooled Data: Phase Transitions of Message Passing; 2017 IEEE International Symposium on Information Theory, ISIT) and of interest for ML/NN community (e.g., Jonathan Scarlett and Volkan Cevher; Phase Transitions in the Pooled Data Problem;  31st Conference on Neural Information Processing Systems, NeurIPS 2017).
> It was hard to enlist and contain all such work within a single technical submission,
> but we will add some of these examples to the paper in order to strengthen the motivation and relevance part of our work even further.
>
> 2. **Is the model interesting? I am not completely convinced, for example it is hard to imagine privacy constraints (as suggested by the authors) forcing the intersection size to be exactly one.**
>
> In Section 5.2 we discuss and briefly describe a relatively efficient way to transform our constructions such that they work for even a simpler binary feedback. The most interesting property of our paper is locality, which allows certain range of autonomy to agents representing elements, as now they do not need the whole feedback vector to learn the hidden set (only the part during which they belong to the queries).
> The privacy aspect is implicitly embedded into the locality property. To illustrate it better, consider a set of interactive agents (e.g., servers), each storing some information. Additionally, there is a requester searching for agents that store some specific information, by asking subsets of servers (queries) whether they have it. Only agents which respond positively to the requester get feedback from it. This way, no agent that does not posses the sought information learns anything from the feedback, because it does not respond to queries and so it does not receive any meaningful feedback from the requester. This way the privacy of the group of agents sharing the same information is protected from the other agents during the learning process. On the other hand, LocS and LocAS guarantee that each agent that has the sought information learns which other agents have it (in case of LocAS, agents could even learn only clusters they belong to, each cluster containing potentially different piece of information).
> We will add more intuitions to the most recent version of the paper, while we also believe that the locality and avoidance properties could be further explored in the context of other privacy-related applications.

---

> > ### Comment · Reviewer_2oQv · 2023-12-04
> > **Thanks for response**
> >
> > Thank you for the detailed response, which nicely addresses my comments.

---

### Official Review · Reviewer_jXfX · 2023-11-01

**Soundness:** 4 excellent
**Presentation:** 3 good
**Contribution:** 3 good
**Rating:** 8
**Confidence:** 3

**Summary:**

The submission gives constructive non-adaptive protocols for two group testing problems in which only agents in one query obtain information of the outcome of the query (in this sense queries are local). This is motivated by privacy considerations. The basic problem considered is of this setting with the double-threshold feedback function with both thresholds being 1. The second more restrictive variant requires the avoidance of a set of given forbidden clusters in queries.
For both problems the existence of shorter protocols is known, however no construction exists and the contribution of this manuscript is to provide the first constructive shorter-than-trivial protocols. These protocols are longer by a polylogarithmic factor in the size of the universe.
In addition these protocols are complemented by lower bounds which are quite close to those achieved by the previously known non-constructed protocols.

**Strengths:**

Group testing is a research problem that falls well within the scope of ICLR and the variants considered in this manuscript are well motivated in terms of local queries and non-adaptiveness. And albeit very specific the considered feedback function is a natural first starting point for future research.
The technical contributions are well presented and apart from minor grammatical errors I am happy with the quality of the writing.

**Weaknesses:**

There is a gap between the length of the constructed protocols and ones that are known to exist. This is claimed to be very small, which I do not entirely agree with as it contains terms in n. I would definitely consider closing this gap as well as the gaps between lower and upper bounds open questions worth highlighting.
It seems that the upper bound results could be unified into one main result and a corollary and the same holds for the lower bounds. See my question below.

Detailed comments:
- Judging from the last sentence I would expect a more in-depth discussion of how the algorithms could be extended to more feedback functions. Actually this is later only done in some detail for one more type of feedback functions. I would suggest weakening this sentence in the abstract (or omitting it).
- page 1: "before the hidden set" rather than "prior the hidden set"
- page 1: "using as small *a* number"
- page 2: "ask *the* question of efficient"
- page 2: "Intuitively *the* presence"
- page 2: "each consecutive queries" is confusing because normally I would think this means consecutive pairs.
- page 2: I think it would be appropriate to give some bibliographic references for the encoding/decoding methods being a motivation for non-adaptive protocolls.
- page 3: the caligraphic N notation for the universe is not really used and could be omitted.
- page 3: Maybe for completeness it would be good to say that queries are subsets of the universe.
- page 3 (and in other places): I think a source from 2018 is not particularly recent.
- Lemma 1: "For *all* positive integers"
- page 6: "between such" rather than "between in such"
- page 8: "prospective" rather than "perspective"
- page 9: "*The* adaptive version"
- page 9: "*the* context of"

**Questions:**

- Is it true that the result for (n,k)-LocS could have been presented as a corollary of (n,k,l)-LocS? If so, why did you decide against it?

---

> ### Author Response · Authors · 2023-11-23
>
> We thank the Reviewer for support, inspiring observations and several comments that will help us to improve the presentation.
>
> **Answer to Question:**
>
> * **It seems that the upper bound results could be unified into one main result and a corollary and the same holds for the lower bounds.**
> and
> **Is it true that the result for $(n,k)$-LocS could have been presented as a corollary of $(n,k,\ell)$-LocAS? If so, why did you decide against it?**
>
> Thanks for this observation.
> Indeed, one could obtain a $(n,k)$-LocS from $(n,k,\ell)$-LocAS, constructed in Section 4, by setting $\ell=1$. However, the result would have larger polylogarithmic factor, and thus, would be asymptotically less efficient than the $(n,k)$-LocS obtained in direct construction in Section 3. The actual benefit of the construction in Section 3 depends on parameter $k$, for instance, if $k$ is at least some root of $n$, i.e., $k\sim n^c$ for some $c\in (0,1)$, the direct construction of $(n,k)$-LocS gives polylogarithmic factor of $\log^2 n$, as in Theorem 3.1, while obtaining $(n,k)$-LocS from $(n,k,1)$-LocAS construction in Section 4 would have polylogarithmic factor of $\log^3\log\log n$, as in Theorem 4.1 for $\ell=1$.
>
> The main technical difference between the two constructions is that, in order to assure avoidance property, we use more prime numbers in the construction in Section 4 comparing to the one in Section 3. In order to be able to select sufficient number of different prime numbers, we are bounded by the distributions of prime numbers -- each imposing roughly a logarithmic overhead. Possibly, one could improve the $(n,k,\ell)$-LocAS construction by taking *certain distribution of triplets* of prime numbers to induce smaller polylogarithmic overhead than the *product of three distributions* of prime numbers (used in our construction) -- this is, however, a challenging question in number theory, left for future study.
>
> Another benefit of having Section 3 is that the direct construction of $(n,k)$-LocS is slightly less complex than the construction of $(n,k,\ell)$-LocAS in Section 4 - therefore, it gradually introduces the readers first to obtaining the locality and next, adding avoidance property.
>
> We will add this whole discussion to Section 5, changing its title to "Further Discussion and Extensions".
>
> **Addressing Weaknesses and Detail Comments:**
>
> * **There is a gap between the length of the constructed protocols and ones that are known to exist. This is claimed to be very small, which I do not entirely agree with as it contains terms in $n$.**
>
> Indeed, our statement was a bit ambiguous. We have clarified in the new version of the paper what we understand by the fact that the gap is small: the ratio between the two is only a polylogarithmic function of input parameters, where the degrees of polylogarithm are small. We will clearly emphasize that obtaining asymptotically optimal constructions (i.e., with ratio being a constant that does not depend on input parameters) and further improving constant factors are challenging open directions.
>
> * **Judging from the last sentence I would expect a more in-depth discussion of how the algorithms could be extended to more feedback functions. Actually this is later only done in some detail for one more type of feedback functions. I would suggest weakening this sentence in the abstract (or omitting it).**
>
> The Reviewer is right - we will try to be more specific by writing that we discuss only some extensions (in the Section 5), while others could be subject to future work.
>
> * **page 2: "each consecutive queries" is confusing because normally I would think this means consecutive pairs.**
>
> We will re-write this sentence to clarify that in adaptive GT, algorithm can design an $i$-th query based on the feedback received for the preceding $i-1$ queries.
> A non-adaptive GT has to construct one universal sequence of queries and cannot change them during the learning process.
>
> * **page 2: I think it would be appropriate to give some bibliographic references for the encoding/decoding methods being a motivation for non-adaptive protocols.**
>
> We will provide a few citations, including a recent paper by Kowalski \& Pajak (NeurIPS 2022) which clearly separates the query construction algorithm (corresponding to the encoding algorithm) from the set reconstruction algorithm (corresponding to the decoding algorithm).
>
> * **page 1: "before the hidden set" rather than "prior the hidden set" (...)** and other remaining editorial suggestions
>
> We agree with all the suggestions and will update the paper accordingly.

---

> > ### Comment · Reviewer_jXfX · 2023-11-23
> >
> > Thank you for your detailed response to each of my questions and comments. I believe that the changes to the submission outlined in the response add satisfactory clarifications to all points raised in my review.

---

### Official Review · Reviewer_mKAb · 2023-11-01

**Soundness:** 2 fair
**Presentation:** 1 poor
**Contribution:** 2 fair
**Rating:** 3
**Confidence:** 2

**Summary:**

In this paper, the authors provide an algorithm for the following problem: there are two parties where one party tries to discover the elements hidden by the other party by asking queries and analyzing feedback. In their context, the feedback refers to the intersection of the query with the hidden set that they want to learn. In the local algorithm they introduce, they assume that each hidden element is "autonomous", meaning it can analyze the feedback itself for only the queries where this element is a part of. Their goal is to design a deterministic non-adaptive sequence of queries that allows each non-hidden element to learn all other hidden elements.

**Strengths:**

They present how to construct efficient Local Selectors (LocS) and Local Avoiding Selectors (LocAS) in polynomial time. Following the works of (Jurdzinski et al. 2017; 2018), they are the first ones to show how to construct such selectors efficiently. Improving over the works of (Jurdzinski et al. 2017; 2018), since they were only proving the existence of such algorithms.

**Weaknesses:**

- In general, I had a hard time following the writing style of the paper. The sentences are sometimes unnecessarily complicated and/or too long, which makes it harder to understand the ideas presented in the paper. My overall comment is that the paper needs a significant amount of rewriting and polishing.

- The improper citation notation is one of my major concerns about this paper. The existing citations are not inside parentheses which interrupts the flow of the text for the readers.

- There is no need to capitalize Machine Learning in the first paragraph of the introduction.

- As far as I checked, the c.f. usage is wrong throughout the paper (https://blog.apastyle.org/files/apa-latin-abbreviations-table-2.pdf). This document says it is used to provide contrasting information but I observed that it is probably used instead of i.e. or e.g. in this paper.

- This sentence in the introduction needs citations: "Originally GT was applied for identifying infected individuals in large populations using pooled tests..."

- The words in the parentheses can be added to this sentence: "More information, applications and links could be found in the seminal book (of) Du et al. (2000) and recent literature (by) Klonowski et al. (2022); Kowalski & Pajak (2022b)."

- LocAS is mentioned in the footnote of page 3 before it has been introduced in the main body of the text.

- The problem is not motivated enough beyond the group testing setting.

- The frequent use of the dash (-) gives an informal tone to the paper.

- "Intuitively, having elements from other clusters in the query may negatively influence, or even clash, the learning process within a given cluster – hence, the goal is to do local learning of k hidden elements within the cluster and simultaneously avoiding the other ℓ “bad” clusters.": No need for a comma after the clash in this sentence. (Page 3)

- Typo in "qyery". (Page 5)

- Did you mean existing result instead of existential result on Page 7? If not, what is an existential result?

- Typo in "guarantying". (Page 9)

- The beeping feedback mention on Section 5.2 needs citations.

Note: I am aware that almost all of the remarks I make under weaknesses related to language and grammar concerns. I am also ware that some of them are easily fixable. I do not want to discriminate against anybody based on their linguistic abilities. My main problem here is that, when the frequency of such mistakes increase, it makes it harder to focus on the actual contents of the paper. Plus, it signals me that the sufficient effort to express the novelty of the paper clearly and efficiently has not been made. I strongly believe that expressing our ideas clearly is a skill we all should aim to advance ourselves as researchers.

**Questions:**

- What does "Our algorithms could be .. applied to codes." mean? Do you refer to a computer program for empirical implementation? If yes, why not provide it with the paper? Or does it mean that it can be applied to something like error correcting codes? If this is the case, explicit mentioning would be better. (I understand this is being explained on Section 5.3 at the very end but maybe a very brief explanation towards the beginning would have helped the reader better.)

- Are there another applications where learning the hidden set would be useful other than the group testing setting?

- What does "whichever are relevant" in the following sentence mean?: We say that Q is constructible in polynomial time if there exists a polynomial-time algorithm, that given parameters n, k, ℓ (whichever are relevant) outputs an appropriate sequence of queries satisfying the requirements."

- What are existential upper bounds?

---

> ### Author Response · Authors · 2023-11-23
>
> We thank the Reviewer for useful feedback. We hope that our answers would convince the Reviewer to support our paper, which will be thoroughly updated according to the reviews.
>
> **Answers to Questions:**
>
> * **What does "Our algorithms could be .. applied to codes." mean? (...)**
>
> Thank you for pointing this out - throughout the paper, by "code" we mean a coding/decoding function used in information theory, such as error-correcting codes, compression codes, cryptographic codes, etc.
> We will clarify it in the beginning.
>
> * **Are there another applications where learning the hidden set would be useful other than the group testing setting?**
>
> In Section 1 we give references to applications in:
>
> - machine learning, such as: simplifying
> multi-label classifiers, eg. Ubaru et al. (NeurIPS 2020), approximating the nearest neighbor, eg. Engels et al. (NeurIPS 2021),
> or accelerating forward pass of a deep neural network, eg. Liang \& Zou (ISIT 2021);
>
> - stream processing, e.g., extracting the most frequent elements, see Cormode et al. (VLDB 2003),
> Cormode \& Muthukrishnan (ACM ToDS 2005), Cormode \& Hadjieleftheriou (VLDB 2008), Yu et al. (VLDB 2004), Kowalski
> \& Pajak (IJCAI 2022);
>
> - coding, eg. Kautz \& Singleton (IEEE ToIT 1964); Porat \& Rothschild (IEEE ToIT 2011); Cheraghchi \&
> Ribeiro (ISIT 2019);
>
> - network communication, eg. Clementi et al. (SODA 2001), Kowalski \& Pelc (PODC 2003),
> Jurdzinski et al. (PODC 2018).
>
> A part of the cited book by Du \& Hwang \& Hwang (2000) is devoted to applications of GT. See also examples in Kowalski \& Pajak (NeurIPS 2022).
>
> Specific applications close to our exact settings are discussed in more detail in sub-section "Motivation for this work.", including various variants of the problem.
>
> See also Section 5.2 ("Local GT as Codes"), where we discuss applications to coding theory.
>
> There is also a rich literature on GT applications, which we did not include in our work because it studies  slightly different GT settings. Eg., average case instead of worst case performance over some probabilistic distribution of the sets of infected items, random choice of tests with applications in information theory (eg. Alaoui et al. Decoding from Pooled Data: Phase Transitions of Message Passing. ISIT 2017) and of interest for ML/NN community (eg. Scarlett \& Cevher. Phase Transitions in the Pooled Data Problem. NeurIPS 2017).
> We will add some of these examples to the paper in order to strengthen the motivation part of our work even further.
>
> * **What does "whichever are relevant" in the following sentence mean?: "We say that $Q$ is constructible in polynomial time if there exists a polynomial-time algorithm, that given parameters $n, k, \ell$ (whichever are relevant) ..."**
>
> We meant that in the construction of LocS, we are given parameters $n,k$ as input, and the constructing algorithms should work in time polynomial in $n,k$. Whereas the constructions of LocAS, having three input parameters $n,k,\ell$, should work in time polynomial with respect to these three parameters. We will clarify it in the updated version.
>
> * **What are existential upper bounds?**
>
> On p. 7 and in other similar places, we actually meant "existential result".
> By this, or more specifically "existential upper bound", we mean
> that the result, e.g., existence of some type of structure, is formally proven using mathematical arguments, but the proof does not provide an efficient (i.e., working in time polynomial in the size of the input) algorithm for building such a structure. We will provide detail explanation in the paper.
>
> **Addressing weaknesses:**
>
> * **The improper citation notation is one of my major concerns about this paper. The existing citations are not inside parentheses which interrupts the flow of the text for the readers.**
>
> We apologize for this inconvenience, but we followed the ICLR 2024 guidance as in the instruction and sample files:
> https://iclr.cc/Conferences/2024/CallForPapers and https://github.com/ICLR/Master-Template/raw/master/iclr2024.zip .
> We are, however, perfectly fine with changing the style, if needed.
>
> * **This sentence in the introduction needs citations: "Originally GT was applied for identifying infected individuals in large populations using pooled tests..."**
>
> We will add an explicit citation (to the original Dorfman's paper).
>
> * **LocAS is mentioned in the footnote of page 3 before it has been introduced in the main body of the text.**
>
> Thank you for spotting this issue, we will fix it by referring to the the definition of the LocAS provided on the next page.
>
> * **The beeping feedback mention on Section 5.2 needs citations.**
>
> We will add more citations to the beeping model, including the seminal work:
> Cornejo \& Kuhn. Deploying Wireless Networks with Beeps. In: DISC 2010.
>
> * **In general, I had a hard time following the writing style of the paper...**
>
> Thank you for all the spotted write-up problems - we will take them into account when preparing an updated version.

---

### Author Response · Authors · 2023-11-23

We thank the Reviewers for their effort and helpful suggestions. We provided detail answers to questions and concerns individually under each review.

---

### Meta-Review · Area_Chair_ZimE · 2023-12-10

**Metareview:**

The paper studies variants of the group testing problem where we have a hidden set of items from an underlying universe of items, and the goal is to identify this hidden set using queries. The feedback model considered is that, if the query contains exactly one item from the hidden set, then the id of this item is returned. The paper focuses on variants of this problem where the query model satisfies a certain notion of locality, which are motivated by considerations such as privacy and security. The main contributions are polynomial time algorithms that are deterministic and non-adaptive. These are the first constructive results for these problems.

Several of the reviewers appreciated the novelty of the models introduced in this work and the theoretical results provided for them. The reviewers expressed concern regarding the exposition and the fit of this work for a broad machine learning audience. Although the models are theoretically interesting, they are quire specific and the motivation provided is only limited to citing some related references and only brief mentions of possible application domains such as privacy and security. Overall, it is not clear how convincing is the motivation provided and whether this paper is a good fit for a broad machine learning audience.

**Justification For Why Not Higher Score:**

The paper studies variants of the group testing problem under settings that seem quite specific and not sufficiently motivated. It is not clear whether this paper is a good fit for a broad machine learning audience.

**Justification For Why Not Lower Score:**

N/A

---

### Decision · Program_Chairs · 2024-01-16

Reject